# FIRE: Semantic Field of Words Represented as Nonlinear Functions

**Xin Du**
The University of Tokyo
duxin@cl.rcast.u-tokyo.ac.jp

Kumiko Tanaka-Ishii
The University of Tokyo
kumiko@cl.rcast.u-tokyo.ac.jp

## Abstract

State-of-the-art word embeddings presume a linear vector space, but this approach does not easily incorporate the nonlinearity that is necessary to represent polysemy. We thus propose a novel semantic FIeld REpresentation, called FIRE, which is a $D$-dimensional field in which every word is represented as a set of its locations and a nonlinear function covering the field. The strength of a word's relation to another word at a certain location is measured as the function value at that location. With FIRE, compositionality is represented via functional additivity, whereas polysemy is represented via the set of points and the function's multimodality. By implementing FIRE for English and comparing it with previous representation methods via word and sentence similarity tasks, we show that FIRE produces comparable or even better results. In an evaluation of polysemy to predict the number of word senses, FIRE greatly outperformed BERT and Word2vec, providing evidence of how FIRE represents polysemy. The code is available at https://github.com/kduxin/firelang.

## 1 Introduction

Word representations have become an important basic technique in NLP. The most important semantic requirement for a word representation, above all, is that similar words are mapped to similar embeddings. Furthermore, the next two most important properties are *compositionality* [27], i.e., the representation of two composed words matches the composition of the two representations, and *polysemy*, the representation of semantic ambiguity. Intuitively, compositionality is a linear quality [27, 1], whereas polysemy is nonlinear and thus requires a representation via mathematical multimodality, which is difficult to represent within a linear space.

State-of-the-art word representations are based on a linear vector space. The linearity of vectors naturally accommodates compositionality, but the question of how to incorporate polysemy has been an issue. Indeed, recent random-variable representations [29, 23, 25, 18, 10] represent polysemy but at the cost of losing compositionality.

The most important previous work that seems to incorporate polysemy in addition to compositionality could be BERT [8]; however, it is a contextual embedding that represents one vector only under a given context that disambiguates the meaning. In contrast, a polysemous word has multiple meanings even without context. Moreover, disambiguation (e.g., with BERT) is also hard, because semantics is often continuous: the word "bank" has 18 meanings in WordNet [17], which are hard to disambiguate even for humans. Our idea was thus to devise an entirely new representation that is nonlinear but naturally represents linearity as well.

Hence, FIeld REpresentation, which we call FIRE, is a *non-contextual representation* of words. It is realized in a $D$-dimensional space in which every point has a quantity, called a semantic *strength*, that is generated by interactions among words. Every word is represented by a pair consisting of a

set of the word's locations and a nonlinear function $f$ representing the word's context, where the function outputs the semantic strength at a location in the field. Two functions can be linearly added, which provides the basis of compositionality. Polysemy is represented through the set of a word's locations and the multimodality of the function $f$. Through this representation, the similarity of two words is obtained by adding the function values at the word locations.

Our experimental results confirmed that FIRE showed similar or better results in comparison with previous Word2XX representations under settings with the same number of parameters. Furthermore, by representing multimodality over the entire field, FIRE has a much wider expressiveness for polysemy. In an evaluation to predict the number of word senses, FIRE greatly outperformed BERT.

## 2 Related Works

**Vectoral word representations.**    The most widely used word representations are in vectoral forms, such as *Word2Vec* [15]. [12] showed that Word2Vec is essentially a decomposition of the point-wise mutual information matrix. Other vectoral representation methods extract different information from corpora. Important works include *GloVe* [19], *fastText* [6], *ELMo* [20], and *BERT* [8].

It has been attempted to represent polysemy via multiple vectors per word [26], but such a "mixture" of vectors lost the quality of compositionality.

**Gaussian word representations.**    [29] proposed *Word2Gauss*, which represents each word by a Gaussian distribution. They exploited the covariance matrix to model the semantic ambiguity of words. Asymmetric similarity (via the Kullback-Leibler divergence) was further introduced to represent hypernyms and hyponyms. *Word2GM* [3] uses a mixture of (two) Gaussian distributions. It is able to represent polysemy by assigning each Gaussian component to represent a separate meaning of a word. [25] proposed to use the Wasserstein distance between two Gaussian distributions to measure word similarity, and that work was generalized by [18] to use elliptical distributions. Between Gaussian or elliptical distributions, the Wasserstein distance has a closed form and can be computed efficiently. We compare these Word2XX methods in Table 1.

**Multinoulli distributions on point clouds.**    [10] proposed *Word2Cloud*, in which each word is represented by a (discrete) Multinoulli distribution defined on a cloud of 64 points in a low-dimensional space (e.g., $\mathbb{R}^2$). Between Multinoulli distributions, the Wasserstein distance has no closed form, but it is approximated efficiently by the Sinkhorn distance [7]. In the word clouds of Word2Cloud, each point is a Dirac delta function, and we adopt that notion here.

The *context mover's distance* (CMD) [23] also uses the Sinkhorn distance. [23] used pretrained GloVe vectors [19] in $\mathbb{R}^{300}$ to construct a point cloud that represents a "context." In this paper, we do not show quantitative results for CMD, because we were unable to reproduce the original paper's results with the published code. Instead, we report results for Word2Cloud and GloVe: Word2Cloud is theoretically close to CMD, and [23] reported that GloVe showed comparable performance with CMD.

Many of these random-variable word representations incorporate polysemy. However, none except CMD [23] incorporate compositionality. Therefore, they have not been extended to sentence representations. CMD [23] can produce a sentence representation by computing the Wasserstein barycenter of the words, which gives a new Multinoulli distribution on the point cloud. However, that computation is not additive, whereas FIRE is additive.

**Sentence representations.**    A typical sentence representation is a vector obtained as a weighted additive composition of vectoral word representations. Various weighting heuristics have been studied, such as the *term frequency–inverse document frequency* (TF-IDF) [14]. [2] proposed the *smooth inverse frequency* (SIF), a simple weighting scheme that works exceptionally well. In this paper, too, we evaluate the capability of the sentence field representation produced by applying the SIF to FIRE word representation.

State-of-the-art vectoral sentence representation methods adopt an adaptive weighting scheme by using a self-attention mechanism [28]; one implementation includes the sentence representation of BERT [8]. Such methods can also be applied to our word field representation, which remains a future work.

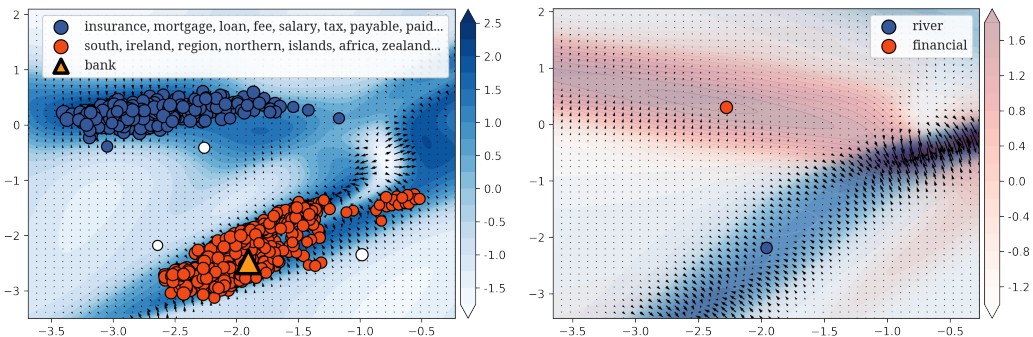

Figure 1: **Left:** Semantic field of *bank* for $D = 2, L = 4, K = 1$, where $D$ is the dimension of this representation space, $L$ is the number of transformation layers (see Formula (7)), and $K$ is the number of polysemy locations (Formula (2)). With $K = 1$, each word has only one location in the field, which is denoted by a scaled Dirac function, $\mu = m\delta(s)$. The triangle shows the location of *bank*. The circles represent the locations of 1000 words that were frequent and similar to *bank*. The words were clustered and colored by using the DBSCAN algorithm [9]. Polysemy is visible via two clusters of words, in finance and geography. The white circles represent the locations of words that DBSCAN considered outliers.
**Right:** Semantic field generated for $D = 2, L = 4, K = 1$ by superposing the functions of *river* (blue) and *financial* (red). The two points show the words' locations. The color intensities indicate the values of the functions at a point, and the arrows indicate the gradients.

## 3   Semantic Field: FIRE

### 3.1   Word Representation

FIRE is a semantic field generated by words. In physics, a field is a space in which every point has some physical quantity, of which a common type is the *strength*, e.g., electric field strength. FIRE uses the semantic strength generated by words. While particles in a physical field are often identical, the particularity of FIRE is that every word produces a different semantic strength and is thus allocated within the field in association with an individual function.

Let $W$ be a set of words and $\mathcal{S}$ be a $D$-dimensional real space. Our semantic field is defined on $\mathcal{S}$. A word $w \in W$ is represented as a 2-tuple $[\mu, f(s)]$, where $\mu$ represents the word's locations. In a previous linear space model [15], a word was usually located at a single point. However, a word can be polysemous, and in such cases, it requires a representation with multiple locations, as suggested in [26]. Let $K$ denote the number of locations. $f(s) : \mathcal{S} \to \mathbb{R}$ is a continuous real-valued function covering $\mathcal{S}$, realized as a neural network. $f(s)$ represents the context generated by the surrounding words of $w$, as will be explained in Section 4.1.

Figure 1 (left) shows an example portion of $\mathcal{S}$ computed for $D = 2$ with $\mathcal{S} = [-4, 4]^2$ and $K = 1$, for the term *bank*. The location $\mu$ for *bank* is shown by the orange triangle. There are two clusters related to *bank*, one for a financial bank and the other for a river bank. The circles show the locations $\mu$ for 1000 words (among the most frequent 50,000 words) that were frequent and similar to *bank*, with *similarity* as defined in the following section.

### 3.2   Similarity of Two Words

In our approach, two words $w_i = [\mu_i, f_i(s)]$ and $w_j = [\mu_j, f_j(s)]$ in a field interact. Given the location $s_j$ of $w_j$, the interaction with $w_i$ is measured by $f_i(s_j)$. As mentioned above, polysemy means that $w_j$ has multiple locations, $s_j^{(k)}, k = 1, \ldots, K$; thus, the $f_i(s_j^{(k)})$ must be added. The location here means that an entity is scattered in a field: it can be multiple and noncontinuous, and it may have different mass at different locations. This notion is mathematically expressed as a *measure*, and an integral with a measure is formulated via the Lebesgue integral.

The simplest measure is the Dirac delta function $\delta(s)$, which indicates that a location is point-wise within a field. For $K = 1$ and $\mu_j \equiv \delta(s_j)$, we have $\int f_i d\mu_j = f_i(s_j)$. By using the Dirac delta function, we can compute the integral exactly without any approximation.

FIRE incorporates polysemy via (1) the multimodality of $f(s)$ and (2) the number $K$ of word locations. As mentioned above, the multimodality of $f(s)$ is implemented via a neural network.

As we also have the symmetric quantity of $w_i$'s location in the field of $f_j(s)$ and $w_j$'s location in the field of $f_i(s)$, the similarity of the two words is naturally expressed as follows:

$$\text{sim}(w_i, w_j) \equiv \int f_i \, d\mu_j + \int f_j \, d\mu_i \quad \forall w_i, w_j \in W. \tag{1}$$

Using a measure for a word's location facilitates a simple representation. For $K \geq 1$, $\mu$ is a Dirac mixture with weights $m^{(k)}$ as follows:

$$\mu \equiv \sum_{k=1}^{K} m^{(k)} \delta(s^{(k)}). \tag{2}$$

By using this $\mu$ for each word, Formula (1) reduces to the following (because $\int f d\delta(s) = f(s)$):

$$\text{sim}(w_i, w_j) = \sum_{k=1}^{K} m_j^{(k)} f_i(s_j^{(k)}) + \sum_{k=1}^{K} m_i^{(k)} f_j(s_i^{(k)}), \tag{3}$$

which is simply a weighted sum of the two functions $f_i(s)$ and $f_j(s)$ at all the locations of $w_j$ and $w_i$, respectively.

The similarity in this work between words and sentences (which we define later in Section 3.3) via a measure is different from the standard notion of distance in a Euclidean space and is related to metric learning methods [11]. Like in metric learning, FIRE is nonlinear with $f$ realized as a multi-layer neural network. Whereas metric learning methods share a large neural network for all inputs, FIRE assigns an individual small neural network to each word.

### 3.3 Composition of Words and Similarity of Two Sentences

A set of $n$ words, $\Gamma = [w_1, \ldots, w_n]$, including sentences, can also be represented in $[\mu, f(s)]$ notation when defined as a weighted average of word representations $\{[\mu_1, f_1(s)], \ldots, [\mu_n, f_n(s)]\}$, as follows:

$$\mu = \sum_{i=1}^{n} \gamma_i \mu_i, \quad f(s) = \sum_{i=1}^{n} \gamma_i f_i(s), \tag{4}$$

where $\gamma_i$ are the weights. In this work, we used SIF weights [2] as the $\gamma_i$. Note that this $f(s)$ is multimodal and thus polysemous, because all the $f_i(s)$ are multimodal.

Figure 1 (right) shows such a superposition of the words *river* (blue) and *financial* (red). Here, *river* is at the lower mid, with $f_{\text{river}}(s)$ forming a diagonal river-like flow from upper right to lower left, whereas *financial* is at the upper left, with $f_{\text{financial}}(s)$ forming a reddish area. The color intensities and arrows represent the function values and gradients, respectively. The field superposing *river* and *financial* is similar to the representation of *bank* on the left, demonstrating the effect of FIRE's linear composition.

The similarity of two sets of words or sentences, $\Gamma = [\mu, f]$ and $\Gamma' = [\mu', f']$, is measured by applying the similarity function (1). Let $n$ and $n'$ denote the respective numbers of words in the two sentences. The words in $\Gamma$ and $\Gamma'$ are assigned weights $\gamma = [\gamma_1, \ldots, \gamma_n]^T$ and $\gamma' = [\gamma'_1, \ldots, \gamma'_{n'}]^T$, respectively. Then, as shown in Appendix B, Formula (1) is simply calculated as follows:

$$\text{sim}(\Gamma, \Gamma') = \gamma^T \Sigma \gamma', \tag{5}$$

where $\Sigma$ is the word similarity matrix, defined as $\Sigma_{ij} = \text{sim}(w_i, w_j)$ with $w_i$ and $w_j$ being words in sentences $\Gamma$ and $\Gamma'$, respectively. In other words, the similarity between two sentences can be obtained from the pairwise word similarities between the sentences.

# 4 Implementation of FIRE

## 4.1 Nonlinear Functions As Planar Transformations

The simplest way to implement the function $f(s)$ would be to construct a neural network NN with a $D$-dimensional input and 1-dimensional output, i.e., $f(s) = \text{NN}(s; \theta)$, where $\theta$ represents the parameters of the neural network. A usual choice for the NN is a *multi-layer perceptron* (MLP).

A more elaborative way to produce the function would be to use the Jacobian of such an NN: $\mathscr{S} \to \mathscr{S}$. The trace of the Jacobian gives the *divergence* at $s$ of the vector field generated by the NN, which is a scalar in $\mathbb{R}$:

$$f(s) \equiv \text{tr}\frac{\partial \text{NN}(s; \theta)}{\partial s}.$$

As the Jacobian $\partial \text{NN}(s; \theta)/\partial s$ can be decomposed into multiplication of the Jacobians of all the NN's layers, it can be computed efficiently. In fact, it was empirically found to work better than the previous simplest MLP.

In particular, a normalizing flow [22], i.e., a reversible neural network, can be used here as NN, as it naturally generates a vector field $\mathscr{S} \to \mathscr{S}$. For our implementation, we adopt a negative multi-layer planar (MLPlanar) transformation [22], which is a simple kind of normalizing flow. Each layer of MLPlanar is determined by two vectors $v$ and $u$ with a scalar $b$, as follows:

$$\text{Planar}(x; \theta) = x + \tanh(v^{\text{T}}x + b)u, \tag{6}$$

where $\theta = \{v, b, u\}$ is the parameter set. [1] The Jacobian of the $l$-th layer with input $s_{(l)}$ is

$$\frac{\partial \text{Planar}(s_{(l)})}{\partial s_{(l)}} = I + A_{(l)},$$

where $A_{(l)} = \tanh'(v_{(l)}^{\text{T}}s_{(l)} + b_{(l)}) \cdot u_{(l)}v_{(l)}^{\text{T}}$ is a rank-one matrix, with the derivative $\tanh' = 1 - \tanh^2$. Thus,

$$f(s) \equiv \text{tr}\frac{-\partial \text{Planar}(s; \theta)}{\partial s} = -\text{tr}\prod_{l=1}^{L}\big(I + A_{(l)}\big), \tag{7}$$

where $s_{(l)} = \text{Planar}(s_{(l-1)})$ and $s_{(0)} = s$. Formula (7) uses the negative trace, so that $f$ has modes that are mostly "peaks" as in Figure 1, rather than "valleys." Because similar words are located where $f$ has higher values, the negative trace drives similar words toward a "peak," which facilitates the separation of a word's multiple meanings. This enables FIRE to discover polysemy. Formula (7) expands to a matrix polynomial of degree $L$, which can represent the multimodality of $f(s)$. For the above case with $D = 2$, evaluating $f(s)$ takes $O(L)$ time.

## 4.2 FIRE Training by Skip-Gram with Negative Sampling

For training FIRE, we adopt the *Skip-gram* algorithm with negative sampling (SGNS) [16], an unsupervised method. More advanced training methods (e.g., *GloVe*) can also be applied, which remains a future work.

In SGNS (one negative word per time), a three-word tuple $(w_i, w_p, w_n)$ is sampled from a corpus. Here, $w_i$ is called the *center* word, $w_p$ is a *positive* word that co-occurs with the center word somewhere in the corpus, and $w_n$ is an arbitrary word, called the *negative* word. We use only one negative word per positive, following the suggestion in [12].

The idea of SGNS is to optimize the parameters in word representations such that $\text{sim}(w_i, w_p)$ is high and $\text{sim}(w_i, w_n)$ is low, following the assumption that words co-occurring in the same neighborhood are more similar. For example, the words *bank* and *mortgage* should have a high similarity because they co-occur frequently.

---

[1]The original planar transformation had additional constraints on the parameter values to preserve reversibility. Because FIRE does not require reversibility, we skip those constraints to speed up our implementation. We thus call Formula (6) a *pseudo*-planar transformation.

Following the implementation in [15], the optimization function with respect to similarity is

$$\min \sum_{w_i, w_p, w_n} \sigma\big(-\mathrm{sim}(w_i, w_p)\big) + \sigma\big(\mathrm{sim}(w_i, w_n)\big), \tag{8}$$

where $\sigma(x) = \log(1 + \exp(x))$ is the *softplus* function. Evaluation of the similarity in Formula (3) between two words has $O(KL)$ time complexity when $D = 2$.

Note that Formula (8) is *not* the *triplet loss* in metric learning. A triplet loss is typically used to evaluate a large neural network shared by all words, whereas in FIRE, each word is represented as an individual small neural network defined by Formula (7).

### 4.3 Experimental Settings for Training

We obtained FIRE representations on two datasets: the large *Wacky* [4] dataset and the small *text8* dataset, which contain about 3 billion and 17 million tokens, respectively.

For both datasets, we converted the text to lowercase and tokenized it with the NLTK toolkit [5]. We followed the settings used in previous works and replaced infrequent words with <unk>. For *Wacky*, infrequent words were those occurring less than 100 times, forming a vocabulary of around 270,000 words; for *text8*, the threshold was 5 times, forming a vocabulary of about 70,000 words.

As mentioned in Section 3.1, we used a 2-dimensional space $\mathcal{S} = [-4, 4]^2$.

For the SGNS hyperparameters, we followed the settings recommended by [15]. We used one negative sample per sample, and the *subsample* rate for $w_i$ was 1e-5 for *Wacky* and 1e-4 for *text8*. The negative sampling probability was adjusted by a power of 0.75. For training FIRE representations, we adopted the *AdamW* optimizer [13]. The *OneCycle* learning rate scheduler [24] was applied, with the maximum learning rate set to 0.005.

As noted above, the time complexity for training is $O(KL)$. For the *Wacky* dataset, all methods were trained for three epochs, which took about 10 hours for FIRE with 50 parameters per word on a single GPU. For *text8*, FIRE was trained for 15 epochs.

In similarity benchmarks, conventional vectoral representations use the cosine similarity function, where $\mathrm{sim}(w_i, w_i) = 1 \; \forall i$, to provide a natural regularization. For fair comparison of FIRE with baselines, we used the following regularization for word similarity in our evaluation:

$$\mathrm{sim}(w_i, w_j) \leftarrow \exp\left( \mathrm{sim}(w_i, w_j) - \int f_i \, \mathrm{d}\mu_i - \int f_j \, \mathrm{d}\mu_j \right). \tag{9}$$

For sentences, substituting $\Sigma$ in Formula (5) with the regularized version of word similarity gives the regularized sentence similarity. Because our similarity function is point-wise, smoothing is also required. Appendix C details this process.

## 5 Baselines

We qualitatively compared FIRE with previous semantic representations, as summarized in Table 1. For fair comparison of such different representations, the most important factor is the number of parameters (sixth column). For FIRE (bottom of the table), $\mu$ requires $K$ locations with $D$ dimensions, plus the number of weights $m^{(k)}$, i.e., $K$, giving $(D+1)K$. As for $f(s)$, $v$ and $u$ (Section 4.1) are $D$-dimensional, are implemented as $L$ layers, and have $L$ intercepts $b$, giving $(2D+1)L$. The total is thus $(2D+1)L + (D+1)K$ parameters per word.

The second column shows whether the method must process a word with a context, and only BERT-large has this requirement. The third and fourth columns in Table 1 indicate whether a method has the qualities of additive compositionality and polysemy; FIRE is the only non-contextual method that theoretically accommodates both. Furthermore, as shown by the fifth column, FIRE and Word2Cloud are the only two that represent polysemy in an interpretable way, i.e., that are visualizable on a 2D plane without dimensionality reduction. However, as will be explained in the next section, Word2Cloud is not comparable with FIRE on word similarity benchmarks.

For an ablation study, we considered a variant of FIRE, denoted as **FIRE/m**, that fixes $m_i^{(k)} = 1$ in Formulas (2) and (3). **FIRE/m** shares the properties of FIRE, but with fewer parameters per word.

Table 1: Qualitative comparison of word representation methods. The second to fifth columns indicate the methods' qualities. The sixth column gives the number of parameters assigned per word: $D$ denotes the dimension of the representation, $K$ is the maximum number of polysemy components, and $L$ is the number of neural layers. The rightmost column gives the time complexity for evaluating the word similarity function, where we consider $D = 2$ for FIRE as a constant.

| method | Non-Contextual | Composi-tionality | Polysemy | Interpre-tability | $N$ (# of parameters) | Complexity $\text{sim}(w_1, w_2)$ |
|---|---|---|---|---|---|---|
| | | | Vectoral representation | | | |
| Word2Vec (2013) | $\checkmark$ | $\checkmark$ | $\times$ | $\times$ | $D$ | $\mathcal{O}(N)$ |
| GloVe (2014) | $\checkmark$ | $\checkmark$ | $\times$ | $\times$ | $D$ | $\mathcal{O}(N)$ |
| BERT-large (2019) | $\times$ | $\checkmark$ | $\checkmark$ | $\times$ | $D = 1024$ | high |
| | | | Random-variable representations | | | |
| Word2Gauss/S (2014) | $\checkmark$ | $\times$ | $\times$ | $\times$ | $D + 1$ | $\mathcal{O}(N)$ |
| Word2Gauss/D (2014) | $\checkmark$ | $\times$ | $\times$ | $\times$ | $2D$ | $\mathcal{O}(N)$ |
| Word2GM/S (2017) | $\checkmark$ | $\times$ | $\checkmark$ | $\times$ | $(D + 2)K$ | $\mathcal{O}(KN)$ |
| Word2GM/D (2017) | $\checkmark$ | $\times$ | $\checkmark$ | $\times$ | $(2D + 1)K$ | $\mathcal{O}(KN)$ |
| Word2Cloud (2019) | $\checkmark$ | $\times$ | $\checkmark$ | $\checkmark$ | $K = 64$ | $\mathcal{O}(N^2)$ |
| CMD (2020) | $\checkmark$ | nonlinear | $\checkmark$ | $\times$ | $K = 200, 400$ | $O(N^2)$ |
| | | | Our semantic-field representations | | | |
| **FIRE (2022)** | $\checkmark$ | $\checkmark$ | $\checkmark$ | $\checkmark$ | $(2D+1)L+(D+1)K$ | $\mathcal{O}(KL)$ |
| **FIRE/m (2022)** | $\checkmark$ | $\checkmark$ | $\checkmark$ | $\checkmark$ | $(2D + 1)L + DK$ | $\mathcal{O}(KL)$ |

We adopted the baselines listed in Table 1 in this work, as explained in Appendix D. Briefly, Word2Gauss and Word2GM represent words as Gaussian mixtures, whereas Word2Cloud represents words as a Multinoulli distribution of $K$ discrete points in a low-dimensional space. For Word2Gauss and Word2GM, "/S" indicates a *spherical* covariance matrix $rI$ requiring one parameter (i.e., $r$), whereas "/D" indicates a *diagonal* covariance matrix $\text{diag}(r_1, \cdots, r_D)$ requiring $D$ parameters. For Word2Cloud, we refer to the results reported in the original paper, because the original code is not available. As Word2Cloud was only implemented on the relatively small *text8* dataset, for comparison, we constructed FIRE models with the same number of parameters and trained them on the *text8* dataset.

## 6  Evaluation on Basic Word Semantics

First, FIRE was compared with the baselines on 12 word similarity benchmarks to test the basic word semantic representational capability. As these benchmarks are for non-contextual methods, BERT-large was not included. For each benchmark, we computed the similarity of a word to other words and generated a ranking of those words. The gold standard was compared via Spearman's correlation coefficient $\rho$, with a higher $\rho$ indicating a better method.

The results are listed in Table 2. Each block groups methods with the same number of parameters. Generally speaking, our method produced comparable or often even better results than the baselines.

For either #par=50 or 100 (third or fourth block, respectively), FIRE was the only method whose performance was comparable to that of the vectoral methods. For example, it achieved an average $\rho$ of 47.9% and 49.8%, respectively, for #par=40 and 50; these $\rho$ values were only 1.1% and 0.6% less than those of Word2Vec (second and third blocks). These results show that our functions and measures have sufficiently large capacity even with a semantic field as small as $[-4, 4]^2$. In Appendix E, we included further experimental results about $K$'s effect. Briefly, our method is computationally efficient even for a larger $K$, because the time complexity of FIRE is $O(K)$, rather than $O(K^2)$ as in Word2GM and Word2Cloud.

Another perspective is to specifically compare the low-dimensional representations, i.e., those with small $D$ (last block). Word2Cloud is such a representation and is comparable to FIRE with $D = 2$. Although Word2Cloud was computed on the smaller *text8* dataset, it produced better similarity scores than the 2-dimensional Word2Vec listed in the top row of Table 2. When trained on *text8*, however, FIRE greatly outperformed Word2Cloud, even with a smaller number of parameters.

Table 2: Spearman's correlation coefficient (%) between the gold standard and the similarity rankings produced by the various word representations. The results are grouped by methods with the same number of parameters. The best and second-best results in each group are in bold and underlined, respectively. The results for the previous methods were obtained by replicating them with the published code, except for Word2Cloud in the bottom block, for which the code was unavailable. The values in the row for Word2Cloud were taken from the original paper [10].

| dataset
# word pairs | | | | | MC
30 | MEN
3000 | RG
65 | WS-s
203 | WS-r
252 | MT
287 | MT
771 | RW
2034 | Verb
143 | YP
130 | SL
999 | SV
3500 | Ave. |
|---|---|---|---|---|---|---|---|---|---|---|---|---|---|---|---|---|---|
| | #par. | $D$ | $L$ | $K$ | | | | | | | | | | | | | |
| **Dataset: Wacky (3B tokens)** | | | | | | | | | | | | | | | | | |
| Word2Vec | 2 | 2 | - | - | 1.6 | 19.4 | 22.2 | 16.1 | 9.3 | 26.6 | 17.4 | 12.6 | -2.2 | 12.0 | -0.2 | 5.5 | 11.7 |
| Word2Vec | 40 | 40 | - | - | 67.8 | **69.1** | 66.1 | **70.1** | **52.1** | **65.0** | **60.7** | **27.8** | 32.6 | **39.6** | 18.4 | 19.3 | **49.0** |
| FIRE/m (ours) | | 2 | 4 | 10 | **72.0** | 65.2 | **71.9** | 64.6 | 47.7 | 63.5 | 60.1 | 25.9 | **32.9** | 31.8 | **20.1** | **19.8** | 47.9 |
| Word2Vec | 50 | 50 | - | - | **72.7** | **70.7** | 69.2 | **71.8** | **54.2** | **65.0** | 62.1 | 27.5 | 30.2 | 41.4 | 19.7 | **20.3** | **50.4** |
| GloVe | | 50 | - | - | 66.8 | 68.7 | 73.4 | 61.3 | 43.2 | 58.6 | **62.2** | **28.0** | 34.3 | 42.1 | 19.4 | 15.8 | 47.8 |
| Word2Gauss/D | | 25 | - | - | 51.7 | 50.2 | 65.5 | 51.1 | 48.0 | 56.9 | 46.0 | 23.3 | 8.8 | 23.9 | 26.0 | 16.4 | 39.0 |
| Word2Gauss/S | | 49 | - | - | 63.7 | 53.7 | 69.8 | 54.3 | 48.6 | 57.4 | 48.9 | 25.9 | 19.5 | 32.8 | **26.9** | 18.6 | 43.3 |
| Word2GM/S | | 23 | - | 2 | 58.4 | 57.7 | 63.7 | 45.8 | 37.4 | 51.0 | 53.6 | 24.1 | 10.0 | 28.3 | 16.0 | 18.4 | 39.8 |
| FIRE (ours) | | 2 | 4 | 10 | 71.8 | 64.9 | **74.2** | 67.5 | **52.1** | 63.0 | 59.2 | 25.8 | **34.5** | **45.5** | 18.7 | 19.8 | 49.8 |
| Word2Vec | 100 | 100 | - | - | 71.4 | **73.6** | 75.2 | **74.2** | **57.7** | **66.4** | **66.3** | 29.1 | **39.8** | 47.1 | 21.5 | **22.9** | **53.8** |
| GloVe | | 100 | - | - | 69.6 | 72.3 | **77.0** | 65.2 | 49.0 | 61.9 | 65.8 | 30.2 | 35.0 | **49.2** | 23.3 | 18.5 | 51.4 |
| Word2Gauss/D | | 50 | - | - | 68.8 | 55.8 | 70.0 | 55.3 | 48.3 | 61.5 | 50.8 | 25.6 | 9.1 | 37.4 | **27.9** | 18.5 | 44.1 |
| Word2Gauss/S | | 99 | - | - | 64.4 | 55.6 | 71.0 | 53.2 | 48.0 | 60.0 | 50.3 | 27.0 | 15.2 | 34.8 | 27.4 | 19.3 | 43.9 |
| Word2GM/S | | 48 | - | 2 | 63.1 | 60.7 | 68.6 | 61.0 | 45.5 | 56.4 | 57.5 | 25.7 | 23.1 | 36.8 | 18.6 | 20.1 | 44.8 |
| FIRE (ours) | | 2 | 8 | 20 | **75.9** | 69.4 | 76.4 | 70.4 | 54.4 | 62.7 | 61.1 | 27.2 | 33.7 | 38.9 | 20.0 | 21.0 | 50.9 |
| Word2GM/S$^b$ | 104 | 50 | - | 2 | 79.1 | 73.6 | 74.5 | 75.5 | 59.5 | 66.6 | 60.8 | 28.6 | 37.4 | 45.1 | 19.7 | 22.6 | 53.6 |
| **Dataset: text8 (17M tokens)** | | | | | | | | | | | | | | | | | |
| Word2Cloud$^a$ | 64 | 2 | - | 64 | 4.0 | 25.0 | 32.0 | 25.0 | 5.0 | 40.0 | 11.0 | 6.0 | 3.0 | - | 9.0 | - | - |
| FIRE (ours) | 50 | 2 | 4 | 10 | 42.6 | 43.2 | 44.8 | 52.4 | 38.3 | **55.6** | 43.0 | 12.3 | 36.5 | 22.8 | **14.3** | 9.4 | 34.6 |
| FIRE (ours) | 64 | 2 | 8 | 8 | 56.2 | 46.3 | 50.6 | 56.9 | 44.3 | 54.8 | 45.7 | 15.8 | 36.8 | 23.5 | 14.2 | 8.7 | 37.8 |

$^a$ The results for *Word2Cloud* were taken from the original paper.

$^b$ The results with #par.=104 was acquired with the best pretrained model available at *https://github.com/benathi/word2gm*, which has a slightly larger vocabulary and was trained with a different scheme.

## 7 Evaluation of Compositionality via Sentence Similarity Benchmark

FIRE's linear compositional capability was evaluated using a sentence similarity benchmark. A sentence representation is produced by composing words in conventional vectoral word representations, making full use of linear compositionality. Similarly, FIRE represents a sentence by composing word representations as described in Section 3.3. The similarity of two sentences is given by Formulas (5) and (9) (see Appendix C), with respect to the SIF weights [2]. We set $\alpha = 10^{-3}$ for SIF.

We compared FIRE with vectoral word representations on the Word-in-Context (WiC) [21] and STS (2012-2016) benchmarks. WiC is a binary classification task asking whether a word's meaning differs in two given context sentences. The semantic similarity scores between sentence pairs were computed, with scores below a threshold indicating that the word's meaning differs in the two sentences. STS evaluates sentence similarity by the Pearson correlation coefficient between the predicted similarity and the gold standard (see Appendix G).

Table 3 lists the results. Note again that Word2Vec and FIRE are non-contextual,

Table 3: Performance on the WiC and STS (2012-2016) benchmarks. See Appendix G for the detailed STS results.

| | $N$ | total par. | WiC | STS |
|---|---|---|---|---|
| Word2Vec | 50 | 13M | 61.8 | 59.7 |
| | 100 | 27M | 62.0 | 61.9 |
| FIRE | 50 | 13M | 61.9 | 57.5 |
| | 100 | 27M | 62.7 | 61.1 |
| BERT-large | 1024 | 340M | 66.5 | 63.3 |

whereas BERT-large is contextual. The second and third columns give the model's number of parameters per word and its total parameters, respectively. The fourth and fifth columns give the accuracy and correlation scores, respectively.

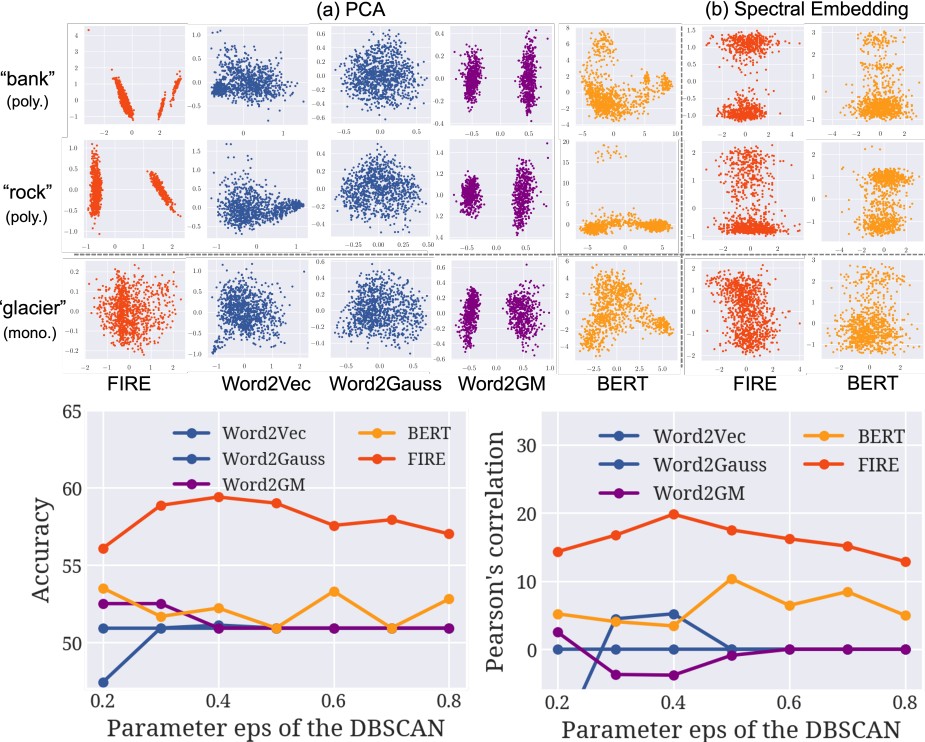

Figure 2: **Upper:** Geometric distributions of word embeddings computed with five methods for the polysemous words *bank* (top row) and *rock* (middle row), and the monosemous word *glacier* (bottom row). **Lower Left:** Accuracy of polysemous/monosemous detection with respect to the DBSCAN threshold parameter "eps." **Lower Right:** Pearson correlation between the detected number of senses and the gold standard, with respect to "eps."

BERT-large, a contextual method, has 20 times more parameters than the non-contextual Word2Vec and FIRE, and its accuracy and correlation were higher because of the context stored in this large memory. Between the non-contextual methods, FIRE performed better than Word2Vec on WiC. BERT's performance was naturally the best because it has context; more importantly, this result demonstrates that FIRE's compositionality is comparable to Word2Vec.

# 8 Evaluation of Polysemy

The quality of FIRE lies in its capability to capture polysemy. Different meanings (i.e., senses) of a word are automatically represented as separate modes of a nonlinear function, as seen for *bank* in Figure 1 with finance- and geography-related modes. Appendix A gives more examples.

We quantitatively evaluated the polysemy of FIRE via the task of *word sense detection*, where the number of word senses is predicted by each method and compared with a gold standard via *WordNet* [17]. From the *Core WordNet* dataset,[2] with 5000 frequent senses of 3739 different words, we selected words annotated with at least three senses (266 words that are strongly polysemous). From Core WordNet and the whole WordNet, we also selected 276 words that are strongly monosemous. We thus used 542 words, with various numbers of word senses. [3]

FIRE was compared with four previous word representations: Word2Vec, Word2Gauss, Word2GM, and BERT. All methods except BERT are non-contextual. With the non-contextual methods, including FIRE, for each target word, the most similar 1,000 words were acquired from the most frequent 50,000 words, and their locations were plotted in the 2D plane. For Word2Gauss and Word2GM, we used the mean vectors of the Gaussian distributions. See Appendix H for the details. In Figure

---

[2] http://wordnetcode.princeton.edu/standoff-files/core-wordnet.txt

[3] The word list is available at https://github.com/kduxin/firelang/data/wordnet-542.txt.

2 (upper), we show the results, with dimensionality reduction by (a) principal component analysis (PCA) and (b) spectral embeddings, for three words: "bank" and "rock" are polysemous, while "glacier" is monosemous.

In contrast, BERT is a contextual model requiring context to count the number of senses. For each target word, 1,000 context sentences were collected from the Wacky dataset, and corresponding BERT vectors for the target word were generated. For a target word that was split into subwords by BERT's default tokenizer, we averaged the subword vectors to represent the target word. The rightmost columns in Figure 2 (upper, a and b) show BERT's results for the three sample words.

As mentioned above, for plotting on the 2D plane, we used two dimension reduction methods, PCA and spectral embeddings, to acquire 2D points from the vectors for BERT (1024-d) and the other methods. For FIRE, the application of PCA is equivalent to performing rotation and normalization, but we nevertheless applied it for fair comparison. For the details, see Appendix H.

Figure 2 (upper) clearly shows how well FIRE captured the different senses. The term "bank" produced three senses, with those to the right corresponding to a river bank and a "large pile," as listed in many dictionaries. On the other hand, such clusters for Word2Vec and BERT are difficult to see. As Word2Vec is monosemous, it is not expected to produce any clusters, while clear clusters are barely visible for BERT. Although BERT performed well in contextual processing, this shows how context and polysemy are not necessarily the same. As for Word2GM, it uses two Gaussian components for each word and is biased to show two clusters, even for the monosemous word "glacier."

For all 542 words, the number of clusters, $t$, was counted with DBSCAN with different thresholds *eps*. For all methods the number was counted on the PCA-reduced 2D planes. For BERT, this PCA result was better than direct application of DBSCAN to the original high-dimensional spaces. The number of clusters was then compared with the gold standard. As the gold standard consisted of 49% ($= 266/542$) polysemous and 51% monosemous words, the result was judged accurate by $t = 1$ for monosemous words and $t > 1$ for polysemous words.

Figure 2 (lower) shows the results for the five methods, with the horizontal axes indicating *eps*. The vertical axes of the lower-left and lower-right graphs show the binary accuracy and Pearson correlation coefficient, respectively. Word2Vec, Word2Gauss, and Word2GM performed at chance level for any *eps* value. BERT was slightly better but could not detect the number of senses. In contrast, FIRE showed a clear capability of identifying polysemy.

## 9 Conclusion

We proposed FIRE, a semantic field representation that is defined on a $D$-dimensional space, with a field of word interactions. A word is represented by a pair that consists of a set of multiple locations for the word and a function covering the semantic field that represents the word's context. The interaction of words is measured via a similarity function, which is naturally defined as the sum of the functions at the word locations. The multiple possible word locations and the function's multimodality account for the quality of polysemy, and the functional addition formally guarantees the quality of additive compositionality.

FIRE is implemented via a planar transformation of a neural network. Multiple images of a 2D field were shown to demonstrate the capability of polysemy representation. FIRE was also evaluated through a large-scale analysis of word and sentence similarity benchmarks, in which it showed comparable or better results than conventional Word2XX representations. Moreover, FIRE greatly outperformed BERT and Word2Vec in evaluating polysemy by well predicting the number of word senses. These results provide evidence of FIRE's capability to incorporate polysemy as a nonlinear quality with compositionality as a linear quality.

Our future work will entail extension of this framework to a more general representation, which will further improve the capability of FIRE.

## Acknowledgements

This work was supported by JST, CREST Grant Number JPMJCR2114, Japan, and by JSPS, KAK-ENHI Grant Numbers JP20K20492 and JP21J11781.

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
