# A  Further Examples of Polysemy Discovery

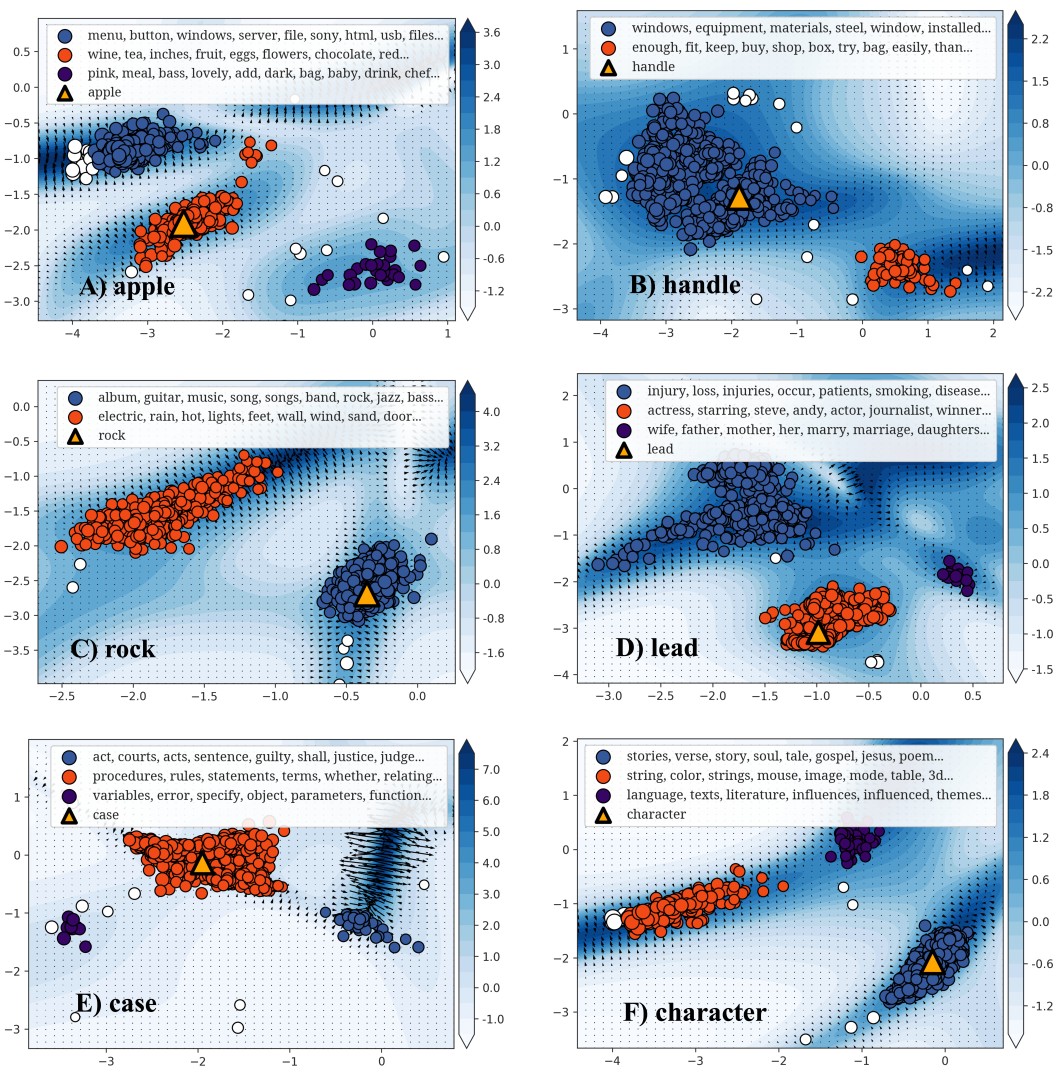

Figure 3: Semantic fields of (A) *apple*, (B) *handle*, (C) *rock*, (D) *lead*, (E) *case*, and (F) *character*, with the most similar words shown at the bottom of each graph. The words were clustered, with corresponding colors for each group, by using the DBSCAN algorithm [9] with eps=0.4, which is based on the Euclidean distance. The white circles represent words that the algorithm evaluated as outliers. In generating the representative words (i.e., those listed in the legends) for each cluster, we used a ranking of the frequency-adjusted word similarity, $\text{sim} \cdot p^{0.3}$, where $p$ is the word's occurring probability in the Wacky corpus, and we adopted the top-ranked words to better show the meaning represented by each cluster.

# B   Derivation of Sentence Similarity

Formula (5) is derived as follows:

$$\begin{aligned}
\text{sim}(\Gamma, \Gamma') &= \int \Big(\sum_{i=1}^{n} \gamma_i f_i\Big) \, \text{d}\Big(\sum_{j=1}^{n'} \gamma_j' \mu_j'\Big) + \int \Big(\sum_{j=1}^{n'} \gamma_j' f_j'\Big) \, \text{d}\Big(\sum_{i=1}^{n} \gamma_i \mu_i\Big) \\
&= \sum_{i=1}^{n} \sum_{j=1}^{n'} \gamma_i \gamma_j' \left( \int f_i \, \text{d}\mu_j' + \int f_j' \, \text{d}\mu_i \right) \\
&= \sum_{i=1}^{n} \sum_{j=1}^{n'} \gamma_i \gamma_j' \text{sim}(w_i, w_j) \\
&= \gamma^{\text{T}} \Sigma \gamma'.
\end{aligned}$$

# C   Smoothing of Similarity Scores by Standardization

One way to improve our framework is to standardize the similarity function. Formally, for a word $w_i$, we apply the following transformation:

$$\overline{\text{sim}}(w_i, w_j) \equiv \frac{\text{sim}(w_i, w_j) - (\text{sim}(w_i, \bar{w}) + \text{sim}(\bar{w}, w_j))/2}{\big(\text{Var}\big[\text{sim}(w_i, w)\big] \text{Var}\big[\text{sim}(w, w_j)\big]\big)^{0.25}}, \tag{10}$$

where the mean is defined as

$$\bar{w} = \frac{1}{|W|} \sum_{w \in W} w = \left[ \frac{1}{W} \sum_{w \in W} \mu_w, \frac{1}{|W|} \sum_{w \in W} f_w \right], \tag{11}$$

the variance is defined as

$$\text{Var}\big[\text{sim}(w_i, w)\big] = \frac{1}{|W| - 1} \sum_{w \in W} \text{sim}(w_i, w - \bar{w})^2, \tag{12}$$

and $\text{Var}\big[\text{sim}(w, w_j)\big]$ is defined similarly.

This standardization should be applied after the regularization given in Formula (9). Thus, in a word similarity benchmark, the predicted word-wise similarity is computed in three steps:

1. compute the word similarity by Formula (1),
2. regularize the similarity score via Formula (9), and
3. apply standardization via Formula (10).

Such standardization works because of our measure's point-wise nature, such that the similarity function in Formula (1) is optimized only at a point. In other words, $f$ is optimized very locally, rather than globally in $\mathcal{S}$. Standardization of the similarity function serves to provide another regularization against overfitting. It is executed after word representations are obtained, and thus, it does not affect the training process.

A typical use of this standardization is for a word similarity benchmark, as in this work. Such a benchmark defines a set of words on which we can perform standardization. We thus define $W$ as this set, and we compute the similarity of word or sentence pairs that are composed of words in $W$.

Similarity benchmarks for sentences also require this standardization procedure, with an additional step (i.e., step 2 in the following) for composing word similarities into sentence similarities:

1. compute the word similarity by Formula (1),
2. compute the sentence similarity by composing the word similarity scores via Formula (5),
3. regularize the sentence similarity via Formula (9), and
4. apply standardization via Formula (10).

## D  Summary of Baseline Methods

**Word2Vec[15]**   [4] A word is represented by a vector, trained via the Skip-gram algorithm.

**GloVe[19]**   [5] This vectoral method is similar to Word2Vec but uses a different training objective function.

**Word2Gauss**   [29][6] A word is represented by a Gaussian random variable, parameterized by a mean vector and a covariance matrix. There are two variants of Word2Gauss: Word2Gauss/S uses a *spherical* covariance matrix $rI$ that requires one parameter (i.e., $r$), whereas Word2Gauss/D uses a *diagonal* covariance matrix $\text{diag}(r_1, \cdots, r_D)$ that requires $D$ parameters.

**Word2GM**   [3][7] A word is represented as a weighted mixture of $K$ Gaussian distributions, each of which is parameterized as in **Word2Gauss**. Word2GM also has two variants: Word2GM/S uses a spherical covariance matrix, and Word2GM/D uses a diagonal covariance matrix.

The original paper recommended setting $K = 2$ and using a spherical covariance matrix. Accordingly, we compared that approach with FIRE in the main text.

**Word2Cloud [10]**   A word is represented by a Multinoulli distribution of $K$ discrete points in a low-dimensional space (e.g., $\mathbb{R}^2$). Word2Cloud also has $K$ components. The similarity between two words is evaluated via the Wasserstein distance, which has at least $O(K^2)$ time complexity even with a fast approximation algorithm. The original paper used $K = 64$.

**BERT-large [8]**   The *large* case-insensitive version of BERT is composed of 24 Transformer layers that are stacked on vectoral word representations. In each Transformer layer, words interact with other words to produce new vectors that contain contextual information. The vectoral output for a word is called a contextualized word vector, which was tested in Section 8.

## E  Supplementary Evaluation Results for Word Similarity

In FIRE, $K$ is the number of locations for each word and is an important factor in polysemy. In comparison, Word2GM also implements polysemy, via mixtures. As seen in Table 1 from the rows for Word2GM and FIRE, the mixture-based approach performed well even with small $D$.

From this perspective, we compared Word2GM and FIRE/m in terms of $K$, under the constraint of $D = 2$. Here, we used FIRE/m for fair comparison, so that we could construct models with a close number of parameters to Word2GM. The number of NN layers, $L$, was also set so that both models would have roughly equal numbers of parameters, as listed in the second to fifth columns of Table 4.

As mentioned in Section 4, our method is computationally efficient even for a larger $K$: the time complexities of Word2GM and Word2Cloud grow by at least $O(K^2)$, whereas FIRE has $O(K)$ complexity. For example, the Word2GM training took around 24 hours for $K = 2$, but the time grew to around 9 days for $K = 6$. Accordingly, we only tested $K = 2, 3, 4, 5$ for Word2GM, whereas for FIRE, we tested $K$ up to 15.

Figure 4 shows the average Spearman's $\rho$ across the tasks of the word similarity benchmark. The values are listed in Table 4 (rightmost column). For Word2GM, an increase in $K$ did not improve the performance for $K \geq 4$. In contrast, FIRE enabled implementation of large $K$ (e.g., 10), for which the performance greatly improved. Thus, FIRE has a larger potential to improve the performance on word similarity benchmarks, by increasing the parameter $K$.

---

[4] https://github.com/dav/word2vec.
[5] https://github.com/stanfordnlp/GloVe.
[6] We used the code for Word2GM and limited it to $K = 1$, because this produced better results than the original code for Word2Gauss at https://github.com/seomoz/word2gauss.
[7] https://github.com/benathi/word2gm.

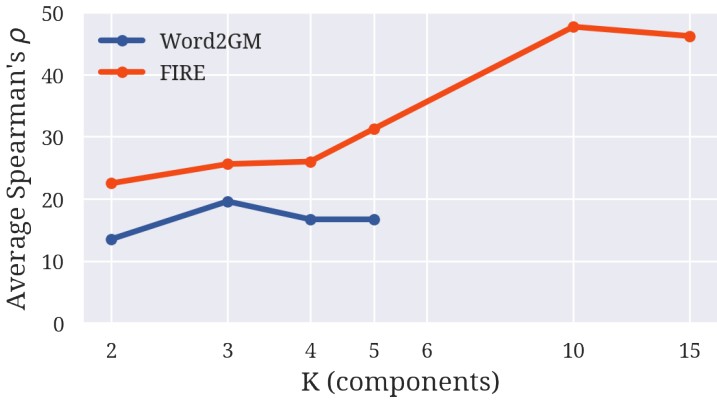

Figure 4: Average Spearman's $\rho$ for Word2GM and FIRE with respect to different $K$ for $D = 2$, with the two methods constrained to have (roughly) the same numbers of parameters for each $K$. Because of its high computational cost, we limited Word2GM to $K \leq 5$.

Table 4: Spearman's correlation coefficient (%) between the similarity rankings produced by the word representations and the gold standard.

| task | | | | | MC | MEN | RG | WS-s | WS-r | MT | MT | RW | Verb | YP | SL | SV | Ave. |
|------|---|---|---|---|----|-----|-----|------|------|-----|------|------|------|-----|-----|------|------|
| # word pairs | | | | | 30 | 3000 | 65 | 203 | 252 | 287 | 771 | 2034 | 143 | 130 | 999 | 3500 | |
| | $D$ | $K$ | #par. | $L$ | | | | | | | | | | | | | |
| Word2GM/S | 2 | 2 | 8 | - | 16.2 | 16.4 | 34.9 | 7.7 | 0.6 | 9.4 | 20.0 | **14.6** | -3.9 | **12.5** | **24.4** | 9.3 | 13.5 |
| FIRE/m | | | 9 | 1 | **23.1** | **30.0** | **45.0** | **38.8** | **20.0** | **36.2** | **30.5** | 14.5 | **11.6** | -2.5 | 12.5 | **11.4** | **22.5** |
| Word2GM/S | 2 | 3 | 12 | - | **34.2** | 22.3 | 37.7 | 18.5 | 7.9 | 19.3 | 24.6 | **16.1** | 6.3 | 12.0 | **24.4** | 11.3 | 19.6 |
| FIRE/m | | | 11 | 1 | 21.0 | **36.6** | **40.0** | **38.8** | **23.3** | **36.3** | **32.5** | 14.0 | **24.0** | **14.1** | 14.7 | **12.3** | **25.6** |
| Word2GM/S | 2 | 4 | 16 | - | 17.8 | 22.0 | 26.7 | 16.8 | 15.3 | 20.3 | 23.5 | **18.4** | -7.2 | **19.6** | **18.3** | 8.5 | 16.7 |
| FIRE/m | | | 13 | 1 | **28.0** | **35.6** | **52.2** | **38.6** | **19.7** | **38.3** | **34.5** | 13.8 | **16.0** | 9.0 | 14.7 | **12.1** | **26.0** |
| Word2GM/S | 2 | 5 | 20 | - | 23.6 | 23.1 | 34.6 | 12.9 | 12.7 | 9.5 | 23.2 | 12.8 | 8.5 | 7.9 | **21.2** | 10.2 | 16.7 |
| FIRE/m | | | 20 | 2 | **25.1** | **48.1** | **49.9** | **40.1** | **28.4** | **44.1** | **42.5** | **17.1** | **32.9** | **16.7** | 18.1 | **12.9** | **31.3** |
| Word2GM/S | 2 | 10 | 40 | - | | | | | (took more than a week) | | | | | | | | |
| FIRE/m | | | 40 | 4 | 71.2 | 66.0 | 71.7 | 63.2 | 53.6 | 64.1 | 59.7 | 23.0 | 30.4 | 29.6 | 21.7 | 18.1 | 47.7 |
| FIRE/m | 2 | 15 | 50 | 4 | 57.1 | 66.6 | 67.4 | 61.9 | 53.3 | 65.5 | 58.4 | 22.7 | 24.2 | 38.6 | 20.1 | 18.3 | 46.2 |

# F   Dimensionalities Other Than 2

Table 5: Average scores on the 12 word similarity benchmarks for FIRE defined on $[-4, 4]^D$ with different $D$. Upper: We set $L = 4$ and $K = 10$ while varying $D$. Lower: We used several parameter sets for $D = 1, 2, 3, 4$, requiring around 50 parameters to represent each word. A larger number of parameters resulted in a better word similarity score.

| $D$ | $L$ | $K$ | # parameters | Average word similarity score |
|-----|-----|-----|--------------|-------------------------------|
| 1 | 4 | 10 | 32 | 39.0 |
| 2 | 4 | 10 | 50 | 49.8 |
| 5 | 4 | 10 | 104 | 50.9 |
| | | | # parameters $\approx 50$ | |
| 1 | 10 | 10 | 50 | 39.4 |
| 2 | 4 | 10 | 50 | 49.8 |
| 3 | 3 | 7 | 49 | 46.0 |
| 4 | 3 | 5 | 52 | 44.9 |

# G Results for Each of 24 STS Datasets

The STS benchmark consists of 24 datasets released during 2012-2016. Each dataset is a list of sentence pairs. For each sentence pair, a human-annotated, gold-standard similarity score is provided. We tested Word2Vec, BERT, and our FIRE on the 24 datasets.

Following the conventional setting, a model's performance was evaluated by the Pearson correlation coefficients between the predicted similarity scores and the gold standard. Spearman's correlation coefficient ($\rho$), which we used for the word similarity benchmarks, was not used for evaluating sentences, because the sentence benchmarks contained many duplicate similarity scores that made Spearman's $\rho$ non-unique.

Table 6 lists the results. Each row is a dataset, and each column is a sentence-embedding model. The bottom row gives the average scores, which are given in Table 3 in the main text.

On average, FIRE performed slightly worse than Word2Vec, though it actually had better performance on many individual datasets.

Table 6: Pearson correlation coefficients on each of the STS datasets (rows) for Word2Vec, BERT, and our proposed FIRE (columns). Each block contains datasets released in the same year.

| $N$ (# parameters) | 50 Word2Vec | 50 FIRE | 100 Word2Vec | 100 FIRE | 1024 BERT-large |
|---|---|---|---|---|---|
| 12/MSRphr | 38.3 | 28.1 | 39.1 | 33.9 | 38.7 |
| 12/MSRvid | 76.0 | 73.9 | 78.8 | 76.3 | 70.6 |
| 12/SMTeuroparl | 46.2 | 33.7 | 47.7 | 36.9 | 50.6 |
| 12/WordNet | 66.7 | 65.0 | 69.3 | 67.7 | 69.0 |
| 12/SMTnews | 54.7 | 35.8 | 51.6 | 48.3 | 53.3 |
| Average | 56.4 | 47.3 | 57.3 | 52.6 | 56.4 |
| 13/FNWN | 39.3 | 37.8 | 42.9 | 42.4 | 43.2 |
| 13/Headlines | 62.2 | 62.7 | 64.2 | 66.7 | 69.8 |
| 13/WordNet | 71.2 | 73.8 | 73.5 | 75.7 | 58.8 |
| Average | 57.6 | 58.1 | 60.2 | 61.6 | 57.3 |
| 14/Forum | 32.6 | 27.3 | 34.9 | 31.9 | 39.9 |
| 14/News | 67.8 | 60.0 | 70.1 | 68.5 | 73.8 |
| 14/Headlines | 58.6 | 58.7 | 60.3 | 62.5 | 66.1 |
| 14/Images | 72.8 | 70.8 | 75.9 | 74.9 | 70.8 |
| 14/WordNet | 74.9 | 76.3 | 77.2 | 78.7 | 68.4 |
| 14/Twitter | 60.8 | 59.2 | 63.8 | 60.6 | 66.1 |
| Average | 61.3 | 58.7 | 63.7 | 62.8 | 64.2 |
| 15/Forums | 49.8 | 52.8 | 54.0 | 54.6 | 54.9 |
| 15/Students | 64.6 | 64.9 | 66.7 | 66.0 | 72.6 |
| 15/Belief | 54.4 | 58.8 | 57.2 | 63.1 | 64.1 |
| 15/Headlines | 66.4 | 66.3 | 69.1 | 69.8 | 71.9 |
| 15/Images | 74.1 | 74.1 | 77.4 | 76.9 | 77.8 |
| Average | 61.8 | 63.4 | 64.9 | 66.1 | 68.3 |
| 16/Answer | 37.1 | 42.5 | 38.0 | 41.4 | 53.0 |
| 16/Headlines | 63.6 | 66.4 | 65.9 | 68.7 | 71.3 |
| 16/Plagiarism | 73.2 | 69.7 | 76.4 | 69.8 | 77.3 |
| 16/Postediting | 67.0 | 67.6 | 69.3 | 71.8 | 81.9 |
| 16/Question | 59.3 | 54.6 | 62.6 | 58.7 | 55.6 |
| Average | 60.0 | 60.2 | 62.4 | 62.1 | 67.8 |
| Average of all | 59.7 | 57.5 | 61.9 | 61.1 | 63.3 |

# H   Plotting on 2D Plane

In Figure 2 (upper), we showed qualitative results for FIRE and the other methods, where the 1000 most similar words out of the most frequent 50000 words were shown on a 2D plane. Here, we describe the two dimensionality reduction methods that we used.

## H.1   PCA

For Figure 2(a), we applied principal component analysis (PCA) to the word "vectors" and reduced the dimensionality to 2 to make the vectors visualizable on a 2D plane. Because all methods are different, we adapted each one to extract representative vectors for PCA, as follows:

**FIRE** ($D = 2, L = 4, K = 1$)   We used the word location $s \in \mathbb{R}^2$ as the vector. Because we used $K = 1$, there was only one such vector for each of the 1000 words. For FIRE, PCA is equivalent to performing rotation and normalization.

**Word2Vec**   The word vector in $\mathbb{R}^{50}$ was used.

**Word2Gauss/S**   The mean vector of the Gaussian distribution in $\mathbb{R}^{49}$ was used.

**Word2GM/S** ($K = 2$)   Each word was represented by two independent Gaussian components, which represented different senses of the word. For a target word such as "bank," we considered 500 components that were similar to the first component of "bank" and 500 that were similar to the second. As with Word2Gauss, we used the mean vector to represent a component. We used the Word2GM/S model listed in Table 2 (third block), which assigns 50 parameters to each word; thus, the mean vectors were in $\mathbb{R}^{23}$.

**BERT-large**   We collected 1000 sentences for each target word from the Wacky dataset. A sentence was input to the BERT model, and a contextualized word vector of the target word was acquired, with 1024 dimensions.

## H.2   Spectral Embedding of Word Similarity Matrix

Another way to visualize the most similar 1000 words was to use the similarity matrix between words. For FIRE, the similarity was computed with Formulas (1) and (9). For BERT, we used the cosine similarity between the contextualized word vectors.

A spectral embedding was acquired by applying SVD decomposition to the similarity matrix and taking only the first two eigenvectors. We used the implementation in the *sklearn.manifold* package.