# OpenReview forum: "FIRE: Semantic Field of Words Represented as Non-Linear Functions"
_NeurIPS.cc/2022/Conference — NeurIPS 2022 Accept_

### Official Review · Reviewer_GMJr · 2022-06-21

**Rating:** 3
**Confidence:** 4
**Soundness:** 3 good
**Presentation:** 3 good
**Contribution:** 2 fair

**Summary:**

This paper proposes an alternative to the standard linear vector spaces that encode word embeddings. The authors propose FIRE (FIeld REpresentation), in which words are represented as a set of locations and a nonlinear function defined on a field. This enables word representations to model the linear nature of compositionality and the nonlinear nature of polysemy. Following existing work on word and sentence representations, the authors propose methods to calculate word/sentence similarity and to compose word/sentence representations. They also present an efficient way to implement the potential function of a word, using the Jacobian of an MLP, and modify the skip-gram negative sampling optimization function to train their model. FIRE is evaluated on word similarity, word disambiguation, and polysemy prediction. Its performance is compared to existing methods, including Word2Vec and contextual BERT embeddings.  The results show that on word similarity and disambiguation tasks, FIRE performs similar to existing methods. On polysemy prediction, FIRE outperforms the baselines, showing that its nonlinear nature allows it to better model polysemy.

**Questions:**

1. Much of your paper is motivated by the claim that compositionality is a linear quality. What is this claim based on? Transformers are currently state-of-the-art for sentence composition and they are nonlinear.

2.  In Table 1, why do you mark Word2Vec as not being effective with small D? One of the persistent advantages of Word2Vec is its low dimensionality.

3. For the experiments with FIRE in Section 8, why do you have to extract and cluster the representations of 1000 similar words? Would it not be possible to obtain a prediction of the number of senses by just using the potential function of a word? To me this would be more convincing proof of FIRE’s value - if the degree of polysemy is encoded in the word representation itself.

**Limitations:**

The authors should consider adding more discussion on the limitations of their model. It should be made more clear in which contexts FIRE could be useful and in which in couldn't.

**Strengths And Weaknesses:**

FIRE is a nice addition to the word and sentence representation literature. The technique is well motivated from a mathematical perspective and quite distinct from existing methods. However, I do not think the results are strong enough to support the introduction of such model complexity, when compared to the simplicity of Word2Vec and the strength of BERT. Furthermore, I think there are stronger baselines that the model should be compared to, especially for sentence encodings. If more baselines are added and more sophisticated analyses of results are performed, I believe this paper could be a strong submission to a computational linguistics venue.

**Strengths:**

1. FIRE is quite novel and an interesting divergence from the standard approach to encoding words and sentences. As the authors point out, there are many advantages to using locations and nonlinear functions to represent words.

2. The authors build their work on mathematically solid foundations, which they explain and lay out well. The similarity and compositionality calculations are not obvious extensions of FIRE - these are good ideas and they are presented clearly. Furthermore, training such a model is challenging. The efficient Jacobian-based training algorithm is another valuable contribution of this paper.

3. The whole paper is well structured and clearly written. The authors manage to concisely present ideas that are sometimes mathematically quite dense.

**Weaknesses:**

1. The results are not very strong and require further analyses. Achieving performance that is similar to Word2Vec and BERT is impressive, but not sufficient given the complexity of FIRE. If a new model is proposed that is much more complex than existing methods, that complexity must be justified through convincing results. The results seem to suggest little gain over existing methods. I would suggest more experiments that focus on polysemy, since this seems to be the strength of FIRE.

2. More analysis should be done to interpret and investigate the results. If performance metrics alone are not convincing, a good case can still be made for FIRE based on analyses that demonstrate how the model is effectively modeling certain aspects of language. For example, Figure 2 shows that FIRE models polysemy well. More such results should be reported. I suspect FIRE is also advantageous from an interpretability perspective e.g. perhaps it is possible to extract different senses from FIRE.

3. More baselines, and stronger baselines, should be included in the results. On the word-level tasks, techniques like GloVe and FastText would be useful for comparison. On the sentence-level tasks, FIRE should be compared to state-of-the-art sentence embedders like InferSent, SentenceBERT, and Universal Sentence Encoder, as well as different word embedding composition techniques, besides weighted summation (e.g. element wise multiplication, averaging, matrix multiplication). I understand that these methods are contextual, but they are still the benchmark against which new sentence embedding methods should be compared.

---

> ### Author Response · Authors · 2022-08-02
> **Author response to Reviewer GMJr**
>
> We thank you for your comments. Following all the reviewers' comments, we uploaded a new version of the paper with improved figures and tables. The changed parts are in blue, and we would appreciate if you could kindly have a look.
>
> **Weakness: The results are not very strong and require further analyses. I would suggest more experiments that focus on polysemy, since this seems to be the strength of FIRE.**
>
> We improved the qualitative comparison in Table 1, including the complexity, which didn't increase with FIRE.
>
> Furthermore, following your advice, we added further comparisons including polysemy for all experiments, to support the capability of FIRE as shown originally.
>
> **Weakness 2. More analysis should be done to interpret and investigate the results. FIRE is also advantageous from an interpretability perspective.**
>
> We added more analysis and comparison with other methods in Tables 2 and 3 and Fig. 2. As there is no standard polysemy benchmark, what we present is the best analysis we can offer. We agree completely that we need such a benchmark, and that will require a new paper.
>
> Regarding FIRE’s strength for polysemy, we already included Appendix A. As you kindly pointed out, FIRE is interpretable, and different point clouds extract different senses. FIRE is advantageous for visualization and performance evaluation via such figures. We added a new column of “Interpretable Polysemy” in Table 1. We consider FIRE to be the only one of the tested methods whose results are interpretable.
>
>
> **Weakness 3:  More baselines, and stronger baselines, should be included in the results.**
>
> Thank you for this excellent comment. For word-level tasks, we did test GloVe before submission, but it did not beat the Word2Vec method on our dataset, and we thus omitted it. However, given your comment, we included its word similarity results in Table 1.
>
> As for sentence-level performance, we added further evidence on the STS (2012-2016) benchmark in Table 3 and Appendix H. FIRE achieved an average score of 61.1 on 24 STS test sets, which was close to Word2Vec at 61.9. Please note, however, that FIRE is a non-contextual method, whereas BERT and the other methods you raised are contextual. Comparison of FIRE and BERT thus unfairly favors BERT-related methods; nevertheless, FIRE competes very well with BERT.
>
> Thank you for providing us with this large to-do list.
>
> **Question: Much of your paper is motivated by the claim that compositionality is a linear quality. What is this claim based on? Transformers are currently state-of-the-art for sentence composition and they are nonlinear.**
>
> Our claim of linearity is primarily based on three previous works: Word2Vec, by Mikolov et al. (2013); the theoretical work by Tian et al. (2017), which revealed the linear nature underlying a combination of word co-occurrence information; and the linear algebraic structure of word senses, by Arora et al. (2018).
>
> Transformers are nonlinear globally, but there is a work on a “linear transformer” [4] that showed performance comparable to the original transformer. On the other hand, our method is essentially nonlinear. Nevertheless, we thank you for your advice, and we would like to better clarify our motivation in the final version.
>
> References
> * [1] Mikolov, Tomáš, Wen-tau Yih, and Geoffrey Zweig. "Linguistic regularities in continuous space word representations." NAACL. 2013.
> * [2] Tian, Ran, Naoaki Okazaki, and Kentaro Inui. "The mechanism of additive composition." Machine Learning. 2017.
> * [3] Arora, Sanjeev, et al. "Linear algebraic structure of word senses, with applications to polysemy." TACL 2018.
> * [4] Katharopoulos, Angelos, et al. "Transformers are RNNs: Fast autoregressive transformers with linear attention." ICML 2020.
>
>
> **Question: In Table 1, why do you mark Word2Vec as not being effective with small D? One of the persistent advantages of Word2Vec is its low dimensionality.**
>
> In our paper, D refers to the dimensionality of the semantic space, not the number of parameters for each word. For Word2Vec, these are the same. The “small-D” property decides whether polysemy can be naturally represented with a low dimensionality to facilitate visualization. We clarified the meaning of “small D” in the new version.
>
> **Question: Would it not be possible to obtain a prediction of the number of senses by just using the potential function of a word?**
>
> We adopted the 1000-point approach because we had to compare all methods. In particular, BERT represents polysemy with concrete contexts (documents), thus requiring contexts to be examined as points. However, as you say, if only for FIRE, clusters could be acquired via only the potential function, which remains for our future work.

---

> > ### Comment · Reviewer_GMJr · 2022-08-03
> > **Reviewer discussion**
> >
> > Thanks for addressing my questions and concerns. The changes you have made have definitely improved the paper. In its current form it would be a strong submission to a conference focused on NLP / computational linguistics. However, I still think that it is not suitable for this venue. The results are not strong enough and I do not see the technical contributions being used or adapted in future research outside of exploratory computational linguistics.
> >
> > Most of the results still show too little gain over existing, simpler methods. On word semantics, Word2Vec mostly outperforms FIRE (or achieves equivalent scores). On compositional tasks, I understand that FIRE is at a disadvantage when compared to contextual methods like BERT. But being non-contextual should be viewed as one of the downsides of FIRE, because it means that on some tasks (that require composition or contextual information) it will compare unfavourably to models like BERT, InferSent, SentenceBERT, and Universal Sentence Encoder.
> >
> > As I said in my review, the polysemy experiments are where FIRE shows the most promise. This is confirmed by the new results and plots. These findings would be particularly valuable for computational linguists. If this paper is ever submitted to a NLP venue, I would advise the authors to shift the focus of the paper to the polysemy results, and to further demonstrate contexts in which the interpretability of FIRE is an advantage over existing methods.

---

> > > ### Author Response · Authors · 2022-08-05
> > > **Author response**
> > >
> > > Thank you for your comment on the changes we made. It seems that your sole concern is whether FIRE achieves better scores than other methods. Because research is not only about improving previous methods, we proposed FIRE as an entirely new idea. It has the good quality, which no previous method has, of representing polysemy while keeping previous representation characteristics.
> > >
> > > Our method is non-contextual, which is an important quality of word representation. This is because how we perceive a word is not only contextual but also non-contextual. Consider any word, such as “cat.” Even without context, we recognize that it produces a meaning.
> > >
> > > BERT does not have this fundamental quality, and it is thus limited to being used only for contextual purposes. We consider that an extension of BERT or Word2Vec cannot be a solution here: we examined many such extensions, which are contextual or non-contextual, but are either non-polysemous or non-compositional. Therefore, we proposed a new framework. FIRE is non-contextual, yet it is already competitive with all previous methods. It also has the potential for extension to contextual methods.
> > >
> > > We are very disappointed to receive a review like this in the prestigious milieu of NeurIPS: a review that disregards a fundamental new idea and focuses only on evaluation scores.

---

### Official Review · Reviewer_awZz · 2022-07-04

**Rating:** 8
**Confidence:** 5
**Soundness:** 4 excellent
**Presentation:** 4 excellent
**Contribution:** 4 excellent

**Summary:**

The authors presented FIRE,  a semantic field representation that is defined on a D-dimensional space, with a field of word interactions. I find the method novel and innovative. Authors also conducted relevant analyzes that solidify their theories. In my opinion, the paper is well written, sounds and flows.

**Questions:**

- I was wondering if the authors have a more strong explanation of table 3. I would really like them to bring more structural explanation rather than the # of parameters
- It would have also been great for the authors to discuss the limitation of their method. For instance, they used (D, K, L) = (2, 4, 1). How well will FIRE perform in higher dimensions and with larger Ks (number of positions/locations) and Ls? This would have made the paper stronger
- Has the authors explored other types of functions for f()? Dirac function is great but what about for instance the Dirichlet function with is a `non-step` function?

**Limitations:**

See Questions

**Strengths And Weaknesses:**

- The paper is well written
- The formulas are well explained, simple, and easy to follow
- The implementation of FIRE is also explicitly described.
- Relevant experiments have been conducted and are convincing of the approach. FIRE captures polysemy (better than BERT and Word2Vec) which is also a good thing.

---

> ### Author Response · Authors · 2022-08-02
> **Author response to Reviewer awZz**
>
> We thank you for your comments and are happy that you liked our paper. Our answers to your excellent questions are below. Following all the reviewers' comments, we uploaded a new version of the paper with improved figures and tables. The changed parts are in blue, and we would appreciate if you could kindly have a look.
>
> **Question: I was wondering if the authors have a more strong explanation of table 3. I would really like them to bring more structural explanation rather than the # of parameters**
>
> We agree, and we broke down the number of parameters. Following another reviewer's comment, we also added results on the STS benchmark, which gave the same message as for WiC.
>
> **Question: It would have also been great for the authors to discuss the limitation of their method. For instance, they used (D, K, L) = (2, 4, 1). How well will FIRE perform in higher dimensions and with larger Ks (number of positions/locations) and Ls? This would have made the paper stronger**
>
> Before submission, we tested D=1,2,3,4,5 and found that D=2 already performed well, as shown in this paper. A larger D gave a better result. Please have a look at the new Appendix G, which explains this.
>
> In summary, for D=5, the result on the word similarity benchmark improved from 49.8 to 50.9. When we limited the number of parameters to 50, however, D=2 worked best; this is why we used D=2 in the paper.
>
> As for larger K, we gave one result in Appendix E to illustrate that FIRE outperforms Word2GM for any K and larger K improves the performance.
>
> **Question: Has the authors explored other types of functions for f()? Dirac function is great but what about for instance the Dirichlet function with is a non-step function?**
>
> Thank you very much for suggesting the idea of using a Dirichlet function. We considered non-step functions such as the Gaussian density. Unfortunately, we have not found a competitive alternative to the Dirac function that enables an exact computation of the integral in the similarity function (Eq. 1). Nevertheless, we would like to explore the possibility of implementing your excellent idea in a future work.

---

> > ### Comment · Reviewer_awZz · 2022-08-06
> > **Replies to authors**
> >
> > Thank you for addressing my concerns. I think you should integrate your last reply to the paper and explain exactly why you chose the Dirac delta function (I assume that’s the one you used).
> >
> > Also it’d be interesting to introduce an interpretability or use cases of your approach in the paper.

---

> > > ### Author Response · Authors · 2022-08-07
> > > **Thank you !**
> > >
> > > Thank you so much for your kind comments. Yes, we will do exactly as you say. For other points that you kindly indicated, we intend to integrate as much as possible. Thank you again for your wonderful advice.

---

### Official Review · Reviewer_J2V2 · 2022-07-11

**Rating:** 6
**Confidence:** 3
**Soundness:** 3 good
**Presentation:** 3 good
**Contribution:** 3 good

**Summary:**

This paper proposes to model polysemy by modeling a word as a tuple
consisting of a set of locations in a field and a nonlinear function.
Compared to other methods for modeling polysemy, this approach also
provides a mechanism for composing word representations. The authors
show that the representations can be meaningfully composed, and that
the method outperforms baselines on word-sense prediction tasks.

**Questions:**

1. Is there any reason that the method is not compared to prior method for polysemous word representations in section 8?
2. Is it possible to cluster words using the proposed similarity function, rather than Euclidean distance, in the DBSCAN algorithm?
3. How is the number of word locations chosen?
4. Can you provide more intuition for the choice of the Jacobian as the similarity function?

**Limitations:**

The authors largely address the limitations of this method. It would be good to add a reference for the number of hours required to train the equivalent baseline methods (lines 184-186).

**Strengths And Weaknesses:**

**Strengths**
1. This paper proposes an interesting and unusual new perspective to
   the task of modeling polysemy. I am not aware of prior work using
   this kind of approach, and I find it to be thought-provoking and
   think it could be of general interest.

2. The paper is written in a straightforward style and is easy to
   read.

3. The experiments illustrate that the resulting embeddings capture
   word similarity, polysemy, and compositionality. In particular, I
   find Figures 1 and 3 to be interesting, illustrative examples of
   the type of representation provided by this approach.

**Weaknesses**
1. I am not entirely convinced by the motivation for this approach.
   The authors write that "The two most important semantic
   requirements for a word representation are compositionality [...]
   and polysemy" (line 15). I think many readers would dispute this
   claim: the most important requirement for a word representation is
   that similar words are mapped to similar embeddings, which is the
   property that enables generalization from small datasets.
   (The method is also evaluated on semantic similarity in section 6.)

   More to the point, it is unclear to me why we would want to compose
   polysemous word representations. While I find the problem to be
   interesting, it is hard for me to imagine a practical setting where
   it would be desirable to compose the representations for "river"
   and "mortgage," or to compose the word "bank" without first disambiguating the intended sense.

2. Evaluating polysemy (section 8): Is there any reason that the
   method is not compared to prior method for polysemous word
   representations?

3. DBSCAN algorithm: The DBSCAN algorithm clusters words using the
   Euclidean distance. This seems somewhat inconsistent with the
   similarity function proposed in this paper. It might be interesting
   to use a clustering method that is based on the proposed similarity
   function, e.g. a spectral clustering algorithm.

4. How is the number of word locations chosen? Compared to
   probabilistic methods, it seems less straightforward to determine
   the number of locations per-word from the data.

**Minor comments**
1. It would be helpful to include additional references
   on fields and potentials, including a general reference and a
   references to any prior work applying these ideas to
   representation learning.
2. The variable L is used in the caption of Figure 1 but not
   introduced in the text until section 4, so that part of the caption
   is difficult to understand on the first reading.
3. Motivation for using the Jacobian: The authors write that using the
   Jacobian of the neural network is a "more efficient way to produce
   the potential function" (line 142). In what sense is it more
   efficient? Is it possible to provide another intuitive
   justification for this choice of similarity function?

---

> ### Author Response · Authors · 2022-08-02
> **Author response to Reviewer J2V2**
>
> We thank you for your comments. Following all the reviewers' comments, we uploaded a new version of the paper with improved figures and tables. The changed parts are in blue, and we would appreciate if you could kindly have a look.
>
> **Weakness 1. The most important requirement for a word representation is that similar words are mapped to similar embeddings; It is unclear why we would want to compose polysemous word representations without first disambiguating the intended sense.**
>
> “The most important requirement for a word representation is that similar words are mapped to similar embeddings.” We totally agree: this claim is such a basic premise that we didn't think to mention it, and it of course precedes compositionality and polysemy. We changed our introduction accordingly. Please note, however, that we still evaluated FIRE with a word semantic benchmark (Table 2).
>
> As for the second part of your comment, from one view, words can indeed be first disambiguated and then composed. From another view, however, it is often hard to “disambiguate,” because semantics is often continuous, and polysemy and compositionality should be represented with the same framework. For example, “bank” has a meaning of a piling, which is close to a river “bank,” but the Oxford dictionary separates these meanings. We follow this second view to represent polysemy and composition with one framework, and we will add this argument in the final version.
>
> **Weakness 2.Q1: Is there any reason that the method is not compared to prior method for polysemous word representations in section 8?**
>
> We performed the experiments before submission, and all methods behaved similarly to Word2Vec, at best, so we chose to only show the results for Word2Vec and BERT. However, given the reviewers’ comments, we reintegrated Word2Gauss and Word2GM in Fig. 2, in addition to other spectral embeddings. In summary, the results show that none of the previous methods worked better than random guessing.
>
> **Weakness 3, Q2: It might be interesting to use a clustering method that is based on the proposed similarity function, e.g. a spectral clustering algorithm.**
>
> We implemented your excellent idea of spectral clustering and added illustrations of the spectral embeddings in Fig. 2. The results were similar to those with PCA.
>
> **Weakness 4, Q3: How is the number of word locations chosen?**
>
> The number of word locations, K, can be given arbitrarily as some maximum number of senses, with a computational complexity that is linear in K. Even with K=1, our potential function could capture polysemy in the form of context, as shown in Fig. 1.
>
> As for the choice of K in our experiment, it was chosen to make the total number of parameters consistent with the other methods. Among different choices with the same number of parameters, we did hyperparameter search on a small portion of the dataset and took the best choice.
>
> **Minor Comments: 1. It would be helpful to include additional references on fields and potentials, including a general reference and a references to any prior work applying these ideas to representation learning.**
>
> In our research, we have not found any reference for representation learning on fields and potentials. While the notion of a potential function is related to energy-based models (EBMs), which provide a framework for generative modeling, our method is about representation learning.
>
> The closest work that discussed representation learning in field physics is “A Quantum Field Theory of Representation Learning” by Bamler et al. (2019). It applied concepts in quantum field theory like “spontaneously broken” to aid representation learning. Unfortunately, it is still too distant from our current work.
>
>
> **Minor Comments: 2. The variable L is used in the caption of Figure 1 but not introduced in the text until section 4.**
>
> We apologize, and we added reference formulas and brief descriptions of D, L, and K in the caption.
>
>
> **Minor Comments: 3, Q2; Motivation for using the Jacobian: In what sense is it more efficient? Is it possible to provide another intuitive justication for this choice of similarity functions.**
>
> Intuitively, the trace of a Jacobian is the divergence of a field in physics. Analytically, the Jacobian of a whole neural network is computed by simple matrix multiplication of Eq. 6, as follows:
> $$J_L = (I+A_1)(I+A_2)…(I+A_L),  …….(\*)$$
> a combination of all layers' outputs.
>
> For an MLP, the corresponding formula is as follows:
> $$J_L = NN_L(…NN_2(NN_1(x))).$$
>
> A simple multiplication is of course more efficient than functional application. The multiplicative Jacobian is also a matrix polynomial to accommodate multimodality.
>
> Furthermore, the Jacobian formula (*) has “highway” connections between $J_L$ and early layers such as $A_1$ and $A_2$. These connections were examined before in ResNet and HighwayNet in image processing, and they showed effectiveness in accelerating the training process of deep neural networks.

---

### Official Review · Reviewer_A5Nv · 2022-07-12

**Rating:** 3
**Confidence:** 4
**Soundness:** 2 fair
**Presentation:** 1 poor
**Contribution:** 2 fair

**Summary:**

The authors proposed FIRE (FIeld REepresentation), in which each word is represented by a pair of points in the embedding space (word sense) and scalar fields in the embedding space (contextual information). Polysemy is naturally represented, and sentence representations can be computed directly from the representations of the constituent words (Eq. 4).

Experiments were conducted on the tasks of word similarity, surrounding context similarity, and estimation of the number of word senses. The proposed method was found to be effective to a certain degree.

**Questions:**

I found both the statement that polysemy is represented by a point cloud (line 34) and the statement that polysemy can be represented by a potential function (e.g., Figure 1). Which is the intent of the authors?

**Limitations:**

This would be one of the limitations if the proposed method does not scale to practical dimensions such as 300 or 768. This is also mentioned by the authors in the checklist.

**Strengths And Weaknesses:**

Strengths

* The attempt to express the meaning of a word as a continuous function on support is both novel and fascinating.

Weaknesses

* Unfortunately, this paper is ill-organized and costly to read.
* The experimental setup is limited, and it is difficult to say that the value of the proposed method has been evaluated. The task used in sentence similarity is not standard (the standard one is STS), and the baselines used are only raw word2vec and BERT. The same problem could be seen in word sense experiments.
* The references and allusions of related work are inconsistent. For example, [21] is one of the most directly related studies because it represents words as a point cloud (polysemy) and proposes a compositional method for sentence representation using barycenter (compositionality; line 64). But this method is not compared in the paper (line 73). In contrast, word2vec [13] also computes sentence representations in terms of barycenter of word vectors and is always compared with the proposed method.

---

> ### Author Response · Authors · 2022-08-02
> **Author response to Reviewer A5Nv**
>
> We thank you for your comments. Following all the reviewers' comments, we uploaded a new version of the paper with improved figures and tables.The changed parts are in blue, and we would appreciate if you could kindly have a look.
>
> **Weaknesses: Unfortunately, this paper is ill-organized and costly to read.**
>
> We improved some presentation issues indicated by other reviewers. Although there were other reviewers who liked our presentation, we sought to improve it as much as possible.
>
> **Weaknesses: The experimental setup is limited, and it is dicult to say that the value of the proposed method has been evaluated. The task used in sentence similarity is not standard (the standard one is STS), and the baselines used are only raw word2vec and BERT. The same problem could be seen in word sense experiments.**
>
> In our original paper at submission time, we also compared our word representation with Word2Gauss and Word2GM, with the latter comparison in Appendix G. Given your comment on sentence similarity, we tested FIRE on the STS benchmark (Section 7) for sentence evaluation. In summary, the experimental results for STS were not different from those for WiC.
>
> As for [21], mentioned in the comment below, we also added a qualitative comparison with CMD in Table 1. CMD has both compositionality and polysemy, but in comparison to our FIRE, it is limited because it cannot accommodate small D and the resulting polysemy is not interpretable.
>
>
> **Weaknesses: The references and allusions of related work are inconsistent. For example, [21] is one of the most directly related studies because it represents words as a point cloud (polysemy) and proposes a compositional method or sentence representation using barycenter (compositionality; line 64). But this method is not compared in the paper (line 73). In contrast, word2vec [13] also computes sentence representations in terms of barycenter of word vectors and is always compared with the proposed method.**
>
> [21] is based on pre-existing word embeddings, which are considered as point clouds, and it presents a method to construct a sentence embedding from them. That work does provide a word semantic benchmark result, but it is based on a word representation from another work. Therefore, we did not include [21] in Table 2.
>
> As for the sentence benchmark (Section 7, Table 3), we tried our best with the code from [21] for STS, where the result for N=300 was 66.4 , as reported in [21]. However, we could not reproduce that result, with the code running very slow because of its O(n2) time complexity. Furthermore, in testing [21] with N=50, the results for STS were very poor at a level of 21%, way below the results for Word2Vec and BERT. For this reason, as well, we did not include [21] in Table 2.
>
> **Question: I found both the statement that polysemy is represented by a point cloud (line 34) and the statement that polysemy can be represented by a potential function (e.g., Figure 1). Which is the intent of the authors?**
>
> Originally, lines 116-117 gave the answer to this question. FIRE represents polysemy by μ, with the number of polysemous senses  represented by K. This is the non-contextual polysemy of a word (i.e., “bank” as a river bank or financial bank).
>
> On the other hand, the potential function f represents the context. The context can be polysemous, too, in that the word “bank” for a financial bank can be used in various ways. Through the integral of the μ function, the potential function’s value counts other words.
>
> Accordingly, our intent lies in representing polysemy in both ways. As there is a slight ambiguity to the term “point cloud,” we avoided using it in our paper. For example, the red and blue clusters in Fig. 1 are each “point clouds.” They indicate the locations of μ for words strongly related to “bank,” which had large similarity scores with “bank.” These "point clouds" do show the polysemy of “bank" via interaction among the words’ representations of μ and f.

---

### Meta-Review · Area_Chair_nfXX · 2022-08-29

**Recommendation:** Accept
**Confidence:** Less certain

**Metareview:**

Two reviewers who suggested that we accept the paper had significant engagement with the authors and that improved the paper quite a bit.  The fourth reviewer also made several suggestions regarding further analysis, which also was incorporated by the authors to a large extent, and it improved the paper.  I disagree with the fourth reviewer that Neurips is not the right audience for this paper, while a CL conference is, and don't agree that it is a reason for rejection.  The other review suggestion a strong reject does not hold too much ground since they are suggesting that the authors compare with more baselines (and they do compare with several reasonable baselines) but ignoring the interesting method that the authors have presented.  Overall, I think the paper is quite interesting even though the results are not extraordinarily SOTA and hence we should give it an audience at Neurips.

**Award:**

No

---

### Decision · Program_Chairs · 2022-09-14

Accept